# Anticonvulsive Effects and Pharmacokinetic Profile of Cannabidiol (CBD) in the Pentylenetetrazol (PTZ) or N-Methyl-D-Aspartate (NMDA) Models of Seizures in Infantile Rats

**DOI:** 10.3390/ijms23010094

**Published:** 2021-12-22

**Authors:** Libor Uttl, Tomáš Hložek, Pavel Mareš, Tomáš Páleníček, Hana Kubová

**Affiliations:** 1Department of Experimental Neurobiology, National Institute of Mental Health, Klecany, Topolová 748, 250 67 Klecany, Czech Republic; libor.kvary@seznam.cz; 2Laboratory of Developmental Epileptology, Institute of Physiology, Czech Academy of Sciences, Vídeňská 1083, 142 20 Prague, Czech Republic; Pavel.Mares@fgu.cas.cz; 3Institute of Forensic Medicine and Toxicology, First Faculty of Medicine, Charles University and General University Hospital in Prague, 121 08 Prague, Czech Republic; Tomas.Hlozek@vfn.cz; 4Department of Psychiatry and Medical Psychology 3FM CU and NIMH, 3rd Faculty of Medicine, Charles University in Prague, Ruská 87, 100 00 Prague, Czech Republic

**Keywords:** cannabidiol, epilepsy, seizures, NMDA, pentylentetrazole, PTZ, immature rats

## Abstract

In spite of use of cannabidiol (CBD), a non-psychoactive cannabinoid, in pediatric patients with epilepsy, preclinical studies on its effects in immature animals are very limited. In the present study we investigated anti-seizure activity of CBD (10 and 60 mg/kg administered intraperitoneally) in two models of chemically induced seizures in infantile (12-days old) rats. Seizures were induced either with pentylenetetrazol (PTZ) or N-methyl-D-aspartate (NMDA). In parallel, brain and plasma levels of CBD and possible motor adverse effects were assessed in the righting reflex and the bar holding tests. CBD was ineffective against NMDA-induced seizures, but in a dose 60 mg/kg abolished the tonic phase of PTZ-induced generalized seizures. Plasma and brain levels of CBD were determined up to 24 h after administration. Peak CBD levels in the brain (996 ± 128 and 5689 ± 150 ng/g after the 10- and 60-mg/kg doses, respectively) were reached 1–2 h after administration and were still detectable 24 h later (120 ± 12 and 904 ± 63 ng/g, respectively). None of the doses negatively affected motor performance within 1 h after administration, but CBD in both doses blocked improvement in the bar holding test with repeated exposure to this task. Taken together, anti-seizure activity of CBD in infantile animals is dose and model dependent, and at therapeutic doses CBD does not cause motor impairment. The potential risk of CBD for motor learning seen in repeated motor tests has to be further examined.

## 1. Introduction

Currently, there is a growing interest in the use of phytocannabinoids obtained from cannabis for medical purposes. The most abundant phytocannabinoids in cannabis sativa plants are the psychoactive Δ9-tetrahydrocannabinol (Δ9-THC) and the nonpsychoactive cannabidiol (CBD). The absence of psychoactive effects significantly simplifies use of CBD for medical purposes [1,2,3]. In particular, its putative antipsychotic, anxiolytic and neuroprotective effects are described [4,5,6,7].

The therapeutic potential of CBD in epilepsy has been known for many years, but major progress in characterizing of its antiseizure effects was made only in the last decade. Both preclinical and clinical studies are bringing convincing data concerning antiseizure effects of CBD. In clinical trials, CBD has been shown to effectively reduce the seizure frequency in patients with Dravet’s syndrome or Lennox-Gastaut syndrome [8,9] and since 2018 it is approved by the U.S. Food and Drug Administration (FDA) for the treatment of these two disorders in children [10].

Anticonvulsant properties of CBD and related compounds were assessed in variety of preclinical tests since early 70s, but former studies were bringing controversial results. Whereas in one study CBD prevented maximal electroshock seizures (MES), no effects in the same model were reported in another [11,12]. Opposing results can be associated with route of administration and inadequate CBD levels, because pharmacokinetic properties of CBD are far from ideal (for rev. [13,14]). More recently, anti-seizure activity of CBD was demonstrated in various in vivo as well as in vitro models (for rev. [15]). Despite the fact that CBD is mainly used to treat pediatric epilepsies, information on its effects in preclinical models of pediatric seizures or epilepsies and on pharmacokinetics in immature laboratory animals are very limited.

The mechanisms responsible for anti-seizure activity of CBD are not fully understood. CBD is “multitarget” drug. It interacts with a large number of receptors and biological systems, many of which have implication in regulation of neuronal excitability. Among them, CBD has been proven to affect G-Protein Coupled Receptors, specifically cannabinoid receptors CB1 and CB2, G-Protein-Coupled Receptors 55 and 18, and opioid receptors μ and δ. In addition, CBD was found to interact with Transient Receptor Potential Vanilloid type 1 (TRPV1) and GABA_A_ receptors and Voltage-Gated Sodium Channel (VGSC) (for rev. [16]). Taken together available data, not mechanisms emerged as prevailing in anti-seizure action of CBD and the primary mechanism may vary across experimental seizure or epilepsy models.

Epidemiologic studies show that seizures and epilepsy affect infants and children more than any other age group [17] and newly onset epilepsies occur most frequently during the first years of life [18]. Thus, clinical data suggest that a developing brain is more prone to generate seizures than an adult one and this was confirmed by many studies in experimental animals. In rodents, the period of maximal seizure susceptibility coincides with the second and third postnatal week of life. It is indeed very difficult to compare level of maturation in immature rodents with this in humans. Based on the timing of growth sprout, synaptogenesis, course of myelination, and maturation of neurotransmitter systems, the overall consensus translates P10-12 rodents to human neonates or early infants [19,20]. In addition, published data demonstrate that EEG activity is up to P10 interrupted by periods of electrical silence corresponding to ”tracé alternant” described in preterm newborns but never seen in full-term human newborns [21]. Thus, based on level of maturation of these developmental milestones, P10-P12 rat pups can be compared with human newborns or infants.

The aim of this study was to determine preclinical safety and efficacy of CBD in infantile rats. Efficacy was assessed in P12 rats in two models of convulsive seizures, induced chemically using either pentylenetetrazole (PTZ) or N-methyl-D-aspartate (NMDA). PTZ-induced seizures represent a standard model for routine screening of potential anti-seizure drugs. In high doses, PTZ induces generalized tonic-clonic seizures in all age groups of rodents including newborns. When administered systemically NMDA causes two types of convulsions: age-specific emprostotonic (flexion) in the first three postnatal weeks and generalized clonic-tonic seizures observed in all age groups [22]. PTZ- and NMDA-induced seizures differ substantially by their pharmacological responsiveness.

To our best knowledge, there is no information available concerning CBD pharmacokinetics in immature rodents. To fill this gap, we used additional groups of animals to determine both serum and brain levels of CBD within 24 h after single dose administration. Possible adverse effects were assessed using two behavioral tests pointed on motor abilities, their development and motor learning.

## 2. Results

### 2.1. Serum and Brain Levels of CBD

We examined changes in serum and brain concentrations of CBD in five different time points (0.5, 1, 2, 4 and 24 h) after administration of CBD in doses of 10 and 60 mg/kg (Figure 1). For each dose and time point, a separate group of animals was used. First, we compared CBD concentration in the brain and serum for each dose. After administration of CBD 10 mg/kg the peak concentration in serum (1655 ± 664 ng/mL as detected using AUC) was reached at the interval 0.5 h and in the brain (996 ± 128 ng/g) at the 1 h interval. Comparison of serum and brain concentrations revealed time point effect F(1.708, 15.80) = 49.48, *p* < 0.0001 and time point x tissue interaction F(4, 47) = 8.578; *p* < 0.0001). Concentration of CBD was significantly higher in the brain tissue compared to serum 1 h (*p* = 0.0352) and 24 h (*p* < 0.0001) after drug administration. Peak levels after administration of CBD 60 mg/kg were 12,903 ± 342 ng/mL in serum and 5689 ± 150 ng/g in the brain, and they were reached 2 h after administration. Serum concentrations were significantly higher than CBD concentrations in the brain starting 0.5 h till 4 h after administration (interval effect F(2.080, 90.24) = 90.2, *p* < 0.0001; tissue effect F(1, 10) = 51.88; *p* < 0.0001 and interval x tissue interaction F(4, 37) = 17.86; *p* < 0.0001).

### 2.2. Effects of CBD on PTZ-Induced Seizures

We first examined the effect of CBD in a model of PTZ seizures (Figure 2). Under control conditions, all animals developed generalized tonic clonic seizures with medial score 5. Administration of CBD affected seizure severity and comparison across treatment groups with Kruskal-Wallis test revealed a significant difference (*H* = 18.91*; p* < 0.0001). Animals treated with CBD 60 mg/kg displayed reduced seizure severity compared to both controls (*p* < 0.0001) and CBD 10 mg/kg group (*p* = 0.0013) (Figure 2A).

Under control conditions mean latency to the generalized seizures (with a score 4–5) was 251 s. Pretreatment with CBD resulted in a significant latency increase across groups (F_2,23_ = 5.730, *p* = 0.0096). At a dose of 60 mg/kg, mean latency increased to 354 s (*p* = 0.0020). The lower dose of CBD (10 mg/kg) tended to increase latency, but difference did not reach level of significance (*p* = 0.0986) (Figure 2C).

Administration of CBD affected pattern and incidence of the tonic phase of generalized seizures. Chi-square analysis revealed a significant difference in the incidence of tonic phase among treatment groups (χ^2^ = 22.13, df = 2, *p* < 0.0001) (Figure 2B). Tonic phase of forelimbs was observed in 100% and hind limbs in 50% of controls, CBD in a dose 60 mg/kg completely blocked tonic phase on both forelimbs and hindlimbs (*p* = 0.0014, Fisher’s exact). In all control animals tonic phase in forelimbs started with tonic flexion that progressed to tonic extension. CBD in a dose of 10 mg/kg suppressed this progression (*p* < 0.0001; Fisher’s exact test) leaving incidence of tonic flexion unchanged (Figure 2D).

Mortality was very low in the 30 min interval after PTZ administration in controls (1 of 10) as well as in CBD animals (0 of 8 in both doses groups) showing no statistically significant differences.

### 2.3. Effects of CBD on NMDA-Induced Seizures

CBD in only tested dose of 60 mg/kg was ineffective in the model of NMDA-induced seizures (Figure 3).

### 2.4. Effects of CBD on Sensorimotor Abilities of Infant Rats

#### 2.4.1. Bar Holding Test

Two-way RM ANOVA revealed significant effects of time after CBD administration (F_2,54_ = 18.14, *p* < 0.0001) and treatment (F_2,54_ = 6.619, *p* < 0.0001). Administration of CBD resulted in significant shortening of time spent on the bar compared to controls in the 3rd test, i.e., 24 h after administration a dose of 10 mg/kg (*p* = 0.0029) as well as 60 mg/kg (*p* = 0.0102). Controls spent on the bar significantly longer time in 24 h-interval (i.e., at P13) compared to time assessed at P12 (0.5 h and 1 h interval; *p* < 0.0001 and *p* > 0.0001, respectively) (Figure 4).

#### 2.4.2. Righting Reflexes

Regardless of treatment and interval after drug administration, all animals returned in their normal upright position in less than 1 s.

## 3. Discussion

The main findings of our study revealed that pre-treatment with CBD resulted in a decrease of seizure severity in PTZ model that was driven by selective blockade of the tonic phase of generalized tonic-clonic seizures. In contrast, CBD did not affect NMDA-induced seizures in infantile, P12 rats. Furthermore, the i.p. administration of CBD resulted in long-lasting plasma and brain levels of CBD that persisted up to 4 h after administration.

Prevailing effects of CBD on tonic (i.e., brainstem-mediated [22]) seizures is consistent with data obtained in adult animals in a model of maximal electroshock [11,23] and with effects of CBD derivative cannabidivarin (CBDV) on PTZ-induced tonic phase of generalized seizures in P10 rats [24]. Specific suppression of tonic phase of PTZ-induced seizures as a major effect of some anti-seizure drugs (e.g., lamotrigine and phenytoin [25], carbamazepine and oxcarbazepine [26], felbamate [27], i.e., drugs effective against MES) (for rev. [28]) was demonstrated in this model.

Systemic administration of high doses of NMDA, used in our study, results in two types of seizures that differ by their pharmacological sensitivity and course of their development. Whereas generalized seizures can be elicited in all age groups of rodents, flexion seizures are highly age-specific and can be reliably induced only in animals younger than three weeks [22]. Because of age-related occurrence of this seizure type and its specific pharmacological sensitivity, flexion seizures were proposed as a model of early life seizures (infantile spasms) for drug screening [29]. Consistently with effects of CBDV [24], CBD was ineffective against flexion seizures. These data contradict results of clinical study showing that administration of CBD results in seizure reduction in patients with epileptic spasms [30]. In this study, however, median age of patients was 8 years, i.e., developmental stage that rather corresponds to more mature, juvenile rodents that are more sensitive to anti-seizure effects of CBD-derivative CBVD [24]. In adult mice CBD was found to diminish NMDA-induced generalized seizures [31], but in our experiments it was ineffective against this seizure type in infantile rats. Conflicting outcomes of our and their study can be explained by different routes of drug administration, because in their experiment both CBD and NMDA were injected intracerebroventricularly and also by different developmental stages of animals.

Differences in anti-seizure activity of CBD in two seizure models used in our study can be related to distinct molecular mechanisms responsible for seizure development. Effects of PTZ are mediated by antagonism of GABAergic transmission [32], whereas NMDA receptors must be involved in the generation of flexion seizures. It is supported not only by our findings that only convulsants acting at this type of receptors are able to induce these seizures [22,33,34], but also by a developmental coincidence of the possibility to induce flexion seizures with overexpression [35] and higher sensitivity of NMDA receptors [36,37]. Pharmacological responsiveness of PTZ-induced generalized seizures and NMDA-induced flexion seizures differs substantially. PTZ-induced generalized seizures, or separately their tonic phase, can be suppressed by many anti-seizure drugs with distinct mechanisms of action (for rev. [38]). In contrast, NMDA-induced flexion seizures represent a model of highly pharmacoresistant seizures and studies using this model documented that only limited number of drugs suppresses these seizures like vigabatrin in P12 animal [39] or valproate in P18 rats [29]. No relation between mechanisms of anti-seizure activity and efficacy in a model of NMDA-induced flexion seizures was reported so far.

Several studies have demonstrated that CBD is well tolerated in pediatric patients and no significant central nervous system side effects, or effects on vital signs or mood, have been reported. Preclinical data on safety of CBD in immature rodents are limited. Administration of CBDV in therapeutic doses to P7 rats leads to increased neuronal death in P7 rats (developmental peak sensitivity to anti-seizure drug (ASD)-induced neuronal apoptosis) [40]. This effect was, however, significantly less robust than effects of other ASD that are commonly used to treat neonatal seizures [41]. Because side effects of ASD are not routinely tested in immature laboratory animals we added simple battery of tests to assess effects of CBD in seizure-suppressing doses on motor performance. Tests were selected according to the level of maturation of the sensorimotor system in infantile animals and differ substantially by their difficulty [42]. Whereas righting reflexes mature shortly after birth and represent relatively a simple reaction that corrects the orientation of the body when it is taken out of its normal upright position, the bar holding test is the most demanding motor test in infantile animals [43].

Performance in these motor tests was not affected by CBD one hour after administration. These data support previous study in adult animals showing that CBD exert seizure protection at non-impairing doses [23]. However, unlike controls, CBD treated animals did not improve their performance in repeated bar holding test performed 24 h after CBD administration. Improvement with repetition is a sign of a motor learning and it is commonly seen in many motor tasks in developing animals [42]. Possible negative effects of CBD on learning abilities have to be however further studied, because of limited number of behavioral tests used in our study.

In line with studies in adult rodents [44], CBD quickly penetrated the brain after intraperitoneal administration. Following administration of 60 mg/kg CBD, T_max_ in the brain was reached after 2 h. Both C_max_ and T_max_ were comparable with those in adult rats following i.p. doses of 120 mg/kg (the active dose in adult rodents) [23,44]. This suggests that lower doses of CBD are sufficient to reach an anti-seizure effective level in the brain in infantile rats. In contrast to adults [44], plasma levels were substantially higher compared to brain levels after administration of 60 mg/kg CBD in infantile rats. This is probably due to immaturity of metabolic pathways responsible for degradation of CBD [45], but pharmacokinetics of CBD and other cannabinoids has to be further studied in developing rodents.

The anti-seizure efficacy of CBD and its derivatives was studied only in limited extent in other age groups of developing rodents, but CBDV was found to be more potent against multiple seizure models in juvenile (P20) compared to infantile (P10) rats [24]. Authors attribute this increase of efficacy to the increase of expression of a CBDV target, the TRPV1 receptor.

## 4. Materials and Methods

### 4.1. Animals

Experiments were performed in 12-day-old (P12) male Wistar albino rats (Institute of Physiology of the Czech Academy of Sciences). The day of birth was defined as day 0. In total, 74 animals were used for this study. Rats were housed in a controlled environment (temperature 22 ± 1 °C, humidity 50–60%, lights on 6 a.m.–6 p.m.) with free access to food and water. On day 5, (birth was defined as day 0), the pups were randomly fostered and each litter was adjusted to 10 males. At postnatal day (PD) 12, the animals were marked for identification and mixed by treatment. Each litter consisted of controls and animals treated with different doses of CBD. To exclude a litter effect, animals in any group were selected from different litters. All procedures involving animals and their care were conducted according the ARRIVE guidelines https://www.nc3rs.org.uk/arrive-guidelines (Accessed on 13 December 2021) in compliance with national (Act No 246/1992 Coll.) and international laws and policies (EU Directive 2010/63/EU for animal experiments and the National Institutes of Health guide for the care and use of Laboratory animals (NIH Publications No. 8023, revised 1978). The Ethical Committee of the Czech Academy of Sciences approved the experimental protocol (Approval No. 15/2018, date of approval 4 June 2018).

### 4.2. Drug Administration and Samples Collection

Cannabidiol (CBD; THC-Pharm GmbH (TH526K, 99.8% purity) was dissolved in 2% solution of Tween 20 (Bio-Rad cat.no 170-6531) in phosphate buffer (0.001M, pH 7.4) and sonified for 60 min at 50 °C. Drug solutions were prepared right before administration. CBD was administered intraperitoneally (i.p.) in doses of 10 and 60 mg/kg.

To determine pharmacokinetic profile of CBD, animals were decapitated under ether anesthesia at five intervals (0.5, 1, 2, 4, 24 h) after CBD administration. Each dose and interval group consisted of 6 animals. Serum and brains were collected, frozen in powdered dry ice and kept at −80 °C until analysis.

### 4.3. Assessment of Plasma and Brain Levels of CBD

Cannabidol was determined by in-house developed and validated and certified GC-MS method (certified by Police Presidium of the CR, ref. no.: PPR-31123-7/CJ-2015-990530/evidence no.: 16/2015). A total of 10 μL of deuterated CBD-d3 (5 ng/μL) internal standard solution was added to each 200 μL sample of serum or brain methanol homogenate (5 mL). For brain analysis, 1g of brain tissue was homogenized in 5 mL of methanol. Homogenized brain samples were frozen at −20 °C in an ethanol bath for 10 min and then centrifuged at 4200 rpm for 2 min. Supernatant (4 mL) was placed in a glass tube and evaporated to 200–300 μL. Serum and brain extracts were diluted with a 4 mL sodium acetate buffer with a pH of 4.0 (0.01 mol/L). Serum and brain CBD were extracted with SPE columns (Bond-ELUT, 130 mg, Agilent Technologies) and eluted with hexan/ethyl acetate (1:4 *v*/*v*) and dried under a nitrogen gas stream in a 400 μL glass insert placed in a 1.5 glass vial. The samples were derivatized with 100 μL of N-Methyl-N-(trimethylsilyl) trifluoroacetamide (MSTFA) for 20 min at 80 °C. Quantification of extracted CBD was performed by gas chromatography-mass spectrometry (GC-MS) (GC7860/5742C MSD, Agilent Technologies, Santa Clara, CA, USA) using electron impact ionization in the selective ion mode (CBD: *m*/*z* 391; CBD-d3: *m*/*z* 394). Calibration curve ranges were prepared by spiking drug-free bovine serum or drug-free rat brain homogenate for serum and brain analysis, respectively, at concentrations of 2–100 ng/mL CBD, (ii) 100–1000 ng/mL CBD. The spiked samples were vortexed and treated identically to the real samples. Real samples were diluted if they exceeded calibration ranges. This analytical method for the determination of CBD in the blood and brain tissue was also used in our previous work [46]. Following a comment [47] on our previous article [46], a reanalysis of the method was performed and a variable conversion of CBD to THC on SPE columns in the range of 0–10% was found. To determine any levels of THC in the measured samples, deuterated THC (THC-3d) was added to the samples and the same calibration series were prepared as for CBD. Therefore, if any THC was detected in samples, these were added to the CBD levels and expressed as CBD. Limits of detection (LOD) and quantification (LOQ) were 1 ng/mL and 2 ng/mL, respectively.

### 4.4. Anticonvulsant Effects of CBD

Two models of chemically induced convulsions—PTZ model and NMDA model—were used to assess anticonvulsant potential of CBD. Because measurement of brain levels of CBD showed that in the brain maximal CBD levels are reached 1–2 h after CBD i.p. administration, animals were pretreated with CBD 60 min before administration of chemoconvulsants. Before the beginning of experiment, a simple randomization was used to assign each rat to a particular treatment group. After CBD administration, pups were returned to their dams. Immediately before administration of chemoconvulsants, animals were placed individually in Plexiglas cages with controlled temperature 34 ± 2 °C. Their behavior was monitored for convulsions for 30 min by an experienced observer blinded to treatment.

#### 4.4.1. PTZ Model

Additional groups of animals were used to assess anticonvulsant effects of CBD in PTZ model. PTZ (Sigma, cat #P6500) was injected subcutaneously in a dose of 100 mg/kg [48]. After PTZ injection, incidence and latency of generalized tonic-clonic seizures (GTCS), starting with short running phase and accompanied by a loss of righting reflexes, was registered. The incidence of tonic phase was registered separately. Pattern of tonic phase on forelimbs was further analyzed and number of animals exhibiting only tonic flexion or tonic flexion followed by extension was registered separately. To assess severity of epileptic phenomena animals were assigned a score for the most severe behavioral characteristics as follows [49]:

0 = no changes

0.5 = abnormal behavior (e.g., automatisms, increased orienting reaction)

1 = isolated myoclonic jerks

2 = atypical or incomplete minimal seizures

3 = minimal seizures

4 = generalized seizures without the tonic phase, GCS

5 = complete generalized tonic-clonic seizures, GTCS

Control group consisted of 10 animals, each CBD treated group of 8 animals. Mortality was registered within a 30 min observation period.

#### 4.4.2. NMDA Model

Additional groups of animals were used to assess effects of CBD in this model. Seizures were induced by N-methyl-D-aspartate (NMDA; Sigma, St. Louis, MO, USA) dissolved in 0.01 PBS, pH 7.4 (60 mg/5 mL) injected intraperitoneally in dose of 60 mg/kg [22]. Controls received physiological saline instead of CBD. In this experiment, CBD was administered only in one dose of 60 mg/kg, which was effective in PTZ-test. The control group consisted of 12 animals, and the CBD 60 mg/kg group consisted of 6 animals.

The following types of motor seizures were evaluated:

Flexion seizures (FS), i.e., whole body flexion, mainly tonic (i.e., emprosthotonus), with a loss of righting reflexes (for details [22,29]).Clonic-tonic seizures (CTS) appeared after flexion seizures. The pattern of generalized clonic-tonic seizures was further analyzed as concerns the presence and distribution of the tonic phase.

In addition to incidence of individual seizure types, latency to seizure onset and lethal outcome were registered up to 30 min after NMDA administration.

### 4.5. Effects of CBD on Motor Performance

Effects of CBD on motor skills and their development were tested using the righting reflex test and the bar holding test. Animals in CBD groups designed for 24 h-interval in pharmacokinetic study (6 animals per each group) were tested to assess acute effects on motor performance and also to detect possible effects of single dose exposure on normal development of motor skills. Control group consisted of 9 animals. Animals were tested in three sessions at three time intervals: for the first time before CBD administration (interval 0 h), than 1 h later (interval 1 h) and for the last time before they were sacrificed i.e., 24 h after CBD administration (interval 24 h).

#### 4.5.1. Righting Reflexes

Pups were placed on their back on a tabletop and held in position for 5 s. The time it takes the pup to return to prone position was measured. In each testing session, the test was always repeated three times and mean time of righting was calculated.

#### 4.5.2. Bar holding

The rats were held so that their forelimbs touched a 25 cm long wooden bar extended between two poles 50 cm high. Latency to fall down was registered up to 120 s.

### 4.6. Statistical Analyses

Sample size was determined in advance based on previous experience and following the principles of the three Rs (Replacement, Reduction and Refinement; https://www.nc3rs.org.uk/the-3rs (Accessed on 13 December 2021). At the beginning of the experiment individual animals were randomly allocated to a particular treatment group. All efforts were made to minimize the number of animals used and their suffering.

Outcome measures and statistical tests were prospectively selected. At the beginning of the study, simple randomization was used to assign each animal to a particular treatment group. Data acquisition and analysis were done blinded to the treatment. Data were analyzed using GraphPad Prism 8 (GraphPad Software, San Diego, CA, USA) software. Using the D’Agostino–Pearson normality test, all data sets were first analyzed to determine whether the values were derived from a Gaussian distribution. Data sets that did not meet strict normality criteria were analyzed using Kruskal–Wallis test. One-way ANOVA and two-way repeated measure ANOVA were used to identify the main effect of CBD. In the presence of a significant main effect but without interaction between factors, simple effects were considered. Whenever a significant interaction was identified, the data were subjected to Tukey’s post-hoc test. Differences between two groups (NMDA model) were analyzed by using Mann–Whitney test, because data sets did not meet strict normality criteria. Incidence of individual seizure phenomena and mortality was compared first with χ^2^ test for trends and subsequently control and individual dose groups were analyzed using Fisher exact test. Latency spent on the bar in the bar holding test was compared using Two-way ANOVA RM, multiple comparison by controlling False Discovery Rate (FDR). Because some values were missing, differences between CBD concentrations in the brain and serum were compared using mixed-effect analysis. Peak concentration of CBD was assessed using Area Under Curve (AUC). *p*-value < 0.05 was required for significance, and *q* < 0.05 was taken as discovery.

## 5. Conclusions

Together, anti-seizure activity of CBD is model-dependent in infantile animals. At seizure-suppressing doses, CBD doesn’t impair motor abilities, but it might impair learning abilities early in development and this potential risk should be further analyzed.

## Figures and Tables

**Figure 1 ijms-23-00094-f001:**
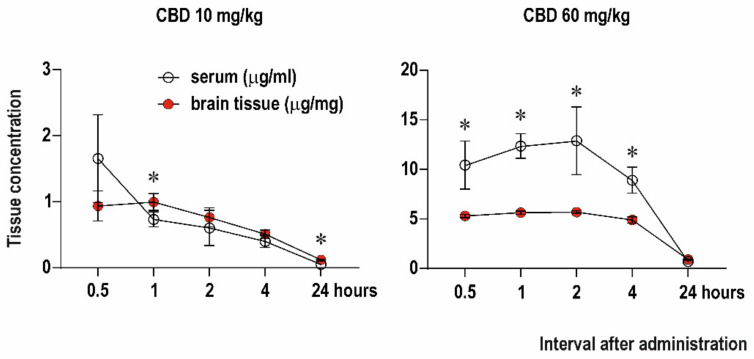
Tissue concentration of CBD. Levels of CBD in serum (μg/mL of plasma, empty circles) and in brain tissue (μg/mg of tissue, red circles) assessed in 5-time intervals after drug administration (0.5–24 h; axis *x*). CBD was administered intraperitoneally in doses 10 mg/kg (left panel) or 60 mg/kg (right panel). Data are expressed as a mean ± SD. Asterisks denote a significant difference between plasma and brain levels.

**Figure 2 ijms-23-00094-f002:**
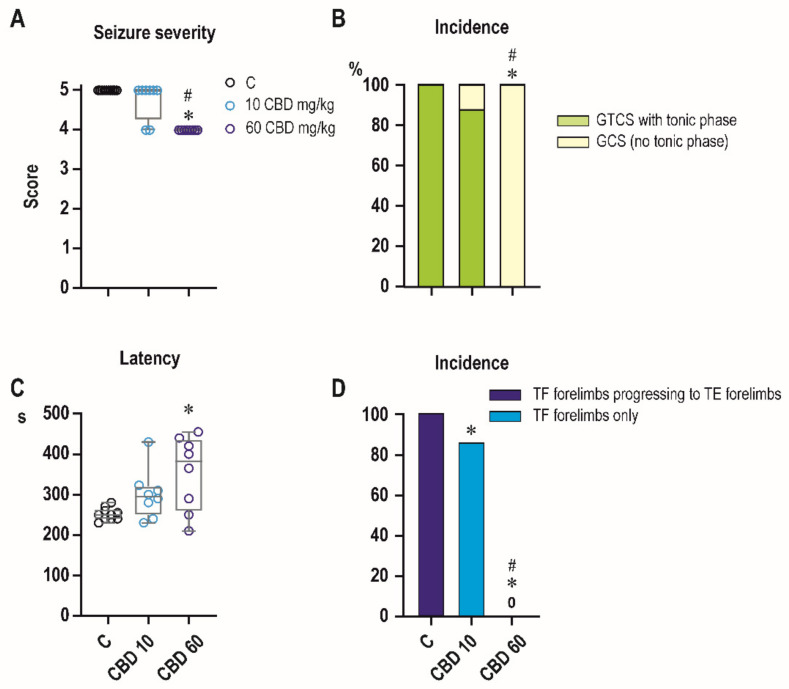
Effects of CBD in PTZ model. Effects of CBD in two doses (10 and 60 mg/kg) on PTZ-induced seizures. CBD in a dose of 60 mg/kg significantly decreased seizure severity (**A**) and this effect was driven mostly by specific suppression of tonic component of generalized tonic-clonic seizures (**B**). The percentage of animals exhibiting complete generalized tonic-clonic seizures (green) and generalized seizures without tonic phase, i.e., generalized clonic seizures (yellow) is on axis y. In this dose, CBD resulted in increase of latency to generalized seizures (**C**). In lower dose of 10 mg/kg, CBD blocked transition from tonic flexion (pale blue; TF forelimbs only) to tonic extension on forelimbs (dark blue; TF forelimbs progressing to TE forelimbs) (**D**). Asterisks denote a significant difference in comparison with controls, # denote a significant difference in comparison between different doses of CBD. *x* axis in all three graphs presents control and two groups with different doses (10 and 60 mg/kg) of CBD. Data in graphs (**A**,**C**) are presented as box plots (the sample median and the first and third quartiles) with whiskers (min and max). Circles mark individual values.

**Figure 3 ijms-23-00094-f003:**
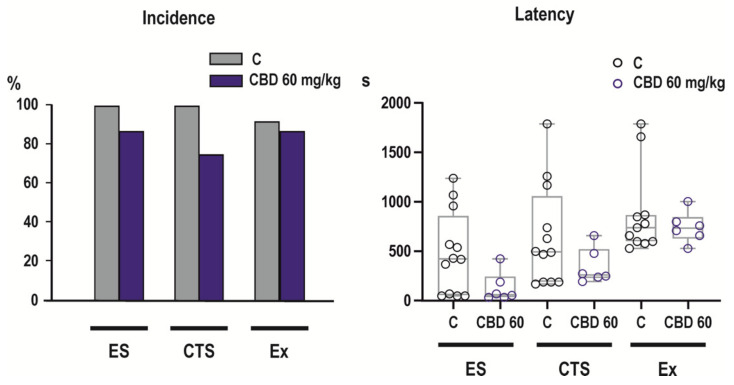
Effects of CBD in NMDA model. Pre-treatment with 60 mg/kg CBD did not affect incidence (left panel) of any evaluated parameters of NMDA-induced seizures—emprostotonic (flexion) seizures (ES), generalized clonic tonic seizures (CTS) or mortality (Ex). Latencies (right panel) of individual parameters did not differ from controls. Other details as in Figure 2.

**Figure 4 ijms-23-00094-f004:**
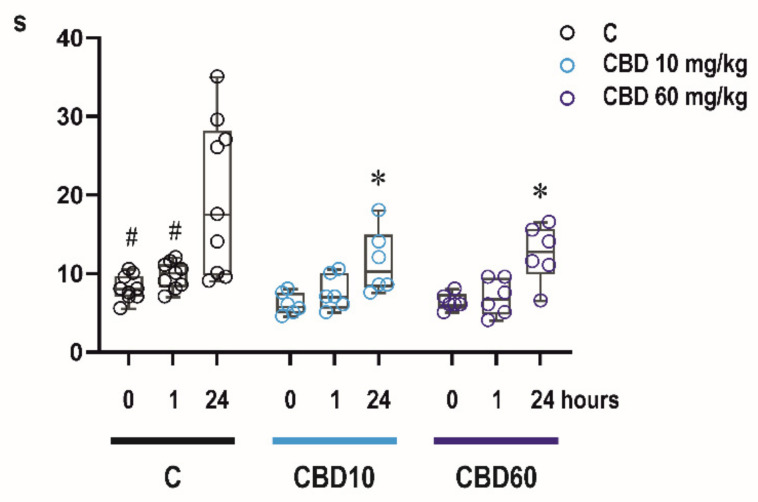
Effects of CBD on motor performance. Bar holding test—Time spent on bar. CBD did not significantly affect time spent on the bar in the bar holding test (axes *y*, in seconds) one hour after drug administration. In controls, time on the bar was significantly longer 24 h after the first exposure to this test. In contrast, animals exposed to either dose of CBD (10 or 60 mg/kg) only tended to improve their performance in this test between the 1st and 3rd test and spent shorter time on the bar compared to controls. For the first time animals were tested before the administration of CBD or solvent. Other tests were performed 1 h and 24 h later. # denotes significant difference between the time spent on the bar in the 3rd and previous tests within one treatment group, * a significant difference in comparison with controls.

## Data Availability

Data are available in the Institute of Physiology, Czech Academy of Sciences, Prague, Czech Republic.

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
