# Peer review of "Anticonvulsive Effects and Pharmacokinetic Profile of Cannabidiol (CBD) in the Pentylenetetrazol (PTZ) or N-Methyl-D-Aspartate (NMDA) Models of Seizures in Infantile Rats"

_ijms, 2021, doi:10.3390/ijms23010094_

Round 1

Reviewer 1 Report

With the typical precision expected from the the senior authors, they report the anti-seizure effect (or lack thereof) of cannabidiol in P12 rats. While CBD dose-dependently suppressed tonic phase of pentylenetetrazole-evoked seizures, it was without effect on NMDA evoked spasms. This is despite appreciable brain concentrations at the time of testing (1 hour after CBD). At the same doses and times, CBD was without effect on motor function (bar holding test, righting reflex), however a mild impact of CBD on motor function in the bar holding test was observed 24 h after treatment.  The inclusion of measurement of CBD levels is a strength, as this allowed the authors to be sure that they were achieving similar plasma concentrations as have been reported to be effective in adult rodents. The statistics are clearly reported, the rationale is solid, the methods sound. Given the growing use and interest in CBD in epilepsy, this paper is excellent, timely, and clearly presented study that will benefit the field.

I have only a couple of minor comment:

  1. can the authors please define TFPK and TEPK (I couldn't figure out the "PK" part of this abbreviation).
  2. to what degree might this response change if the authors tested other ages? (as a point of discussion, not a need for another experiment).

Author Response

Authors would like to thank for helping them to improve the manuscript and correcting mistakes.

1.      We apologize for use of confusing abbreviations. We accidentally left in the manuscript the abbreviation TFPK and TEPK based on the Czech language. PK was replaced with “forelimbs” and it is now explained in the figure legends. 

2.      Anti-seizure effects of CBD were unfortunately not studies in other age groups of rodents. Therefore we added short paragraph based on only available data describing age-related changes in efficacy of CBD derivative, CBDV, with possible mechanisms.

Thank you once again for your kind comments and we hope that we have incorporated them into the manuscript in a suitable manner.

Reviewer 2 Report

After reading the manuscript, my minor concerns are as follows:

  1. Figure 1. Please, replace nanogram units (ng) with microgram (µg) units in both, brain and serum concentrations of CBD in experimental animals. Additionally, please add description to the Y-axis. The results will be better presented in microgram units on Figure 1.
  2. Line 130. The F statistics value - it was really 5,730? Please, replace a comma with a full stop.
  3. Line 222. Please, correct the word “twe”.
  4. Please, add in the Discussion section some more information on molecular mechanisms responsible for the lack of CBD responses to NMDA-induced seizures in immature rats. Such theoretical explanations will be valuable for proper understanding the difference of CBD response in PTZ-induced seizures in rats.

Author Response

We would like to thank to the reviewer for suggestions that helped us to improve and clarify our manuscript.

1.       Typos were corrected as suggested.

2.       In Figure 1 we replaced units (ng with mg) and added description of y-axis as recommended.

In the Discussion we added one paragraph describing differences in molecular mechanisms between PTZ- and NMDA-induced seizures that in our best opinion play a key role in different pharmacological responsiveness of these two models.